# Retaining Mechanical Properties of GMA-Welded Joints of 9%Ni Steel Using Experimentally Produced Matching Ferritic Filler Metal

**DOI:** 10.3390/ma15238538

**Published:** 2022-11-30

**Authors:** Abdel-Monem El-Batahgy, Mohamed Raafat Elkousy, Ahmed Abd Al-Rahman, Andrey Gumenyuk, Michael Rethmeier, Sergej Gook

**Affiliations:** 1Central Metallurgical Research and Development Institute (CMRDI), Cairo 11421, Egypt; 2Metallurgical Engineering Department, Cairo University, Giza 12613, Egypt; 3Federal Institute of Materials Research and Testing (BAM), 12205 Berlin, Germany; 4Institute of Machine Tools and Factory Management, Technische Universität Berlin, 10587 Berlin, Germany; 5Fraunhofer Institute for Production Systems & Design Technology IPK, 10587 Berlin, Germany

**Keywords:** 9%Ni steel, Ni-based austenitic filler metal, matching ferritic filler metal, preheating, post-weld heat treatment, microstructure, mechanical mismatching

## Abstract

Motivated by the loss of tensile strength in 9%Ni steel arc-welded joints performed using commercially available Ni-based austenitic filler metals, the viability of retaining tensile strength using an experimentally produced matching ferritic filler metal was confirmed. Compared to the austenitic Ni-based filler metal (685 MPa), higher tensile strength in gas metal arc (GMA) welded joints was achieved using a ferritic filler metal (749 MPa) due to its microstructure being similar to the base metal (645 MPa). The microstructure of hard martensite resulted in an impact energy of 71 J (−196 °C), which was two times higher than the specified minimum value of ≥34 J. The tensile and impact strength of the welded joint is affected not only by its microstructure, but also by the degree of its mechanical mismatch depending on the type of filler metal. Welds with a harder microstructure and less mechanical mismatch are important for achieving an adequate combination of tensile strength and notched impact strength. This is achievable with the cost-effective ferritic filler metal. A more desirable combination of mechanical properties is guaranteed by applying low preheating temperature (200 °C), which is a more practicable and economical solution compared to the high post-weld heat treatment (PWHT) temperature (580 °C) suggested by other research.

## 1. Introduction

The global need to use natural gas as a clean energy source with lower CO_2_ emissions than petroleum has been increasing as the world seeks to reduce the problem of environmental contamination. Given the current economic and political situation around the world, many countries are concerned about ensuring their energy stability and are storing large quantities of natural gas. However, the biggest gas fields in the world are located far away from the main areas of consumption, so large quantities of gas have to be transported and stored over long distances. Liquefying the gas is of considerable importance for reducing its volume by about 600-times, which obviously simplifies its transportation and storage processes. Transport ships and storage tanks for liquefied natural gas (LNG) are being built around the world [1,2]. Heat-treated ferritic steel alloyed with 9%Ni is widely accepted as the most important material for the fabrication of large LNG storage tanks and vessels where high strength (≥700 MPa) and pronounced notch impact energy (≥34 J at −196 °C/−320 °F) are required [3,4,5]. Welding is commonly used in the manufacturing of LNG transportation and storage equipment; therefore, understanding the factors that affect the quality of welded joints is crucial to maintaining its mechanical properties. Nowadays, LNG equipment is made of 9%Ni steel using arc-based welding processes, mainly gas metal arc welding (GMAW). In this context, the use of austenitic Ni-based consumables has long been prioritized in the LNG sector due to their high toughness and brittle fracture resistance at −196 °C (−320 °F) [6,7,8]. The main limitation of Ni-based fillers is that they do not reach the strength of ferritic steel with 9%Ni. This condition is reflected in design codes for LNG tanks and vessels which base maximum permitted design stresses on the strength of under-matching Ni-based weld joints. Consequently, in order to meet safety requirements and ensure the overall strength of the welded structure, the wall thickness of the LNG units is designed to be excessively thick, which has a negative impact on the economic aspect [9,10,11,12,13,14,15]. In other words, retaining a combination of mechanical properties with this steel type is a major challenge. Autogenous laser beam welding has already been used to produce welded joints with 9%Ni steels that have properties similar to those of the base metal (BM) [16,17,18,19]. It has been demonstrated in many applications that laser welding is an efficient tool for retaining the properties of various difficult-to-weld materials [20,21,22,23,24,25]. However, laser welding of 9%Ni steel is still the subject of comprehensive research and will require several more years to clarify and prove its possible actual applications for LNG facilities manufacturing.

Considering the results of laser beam welding, the production of arc-welded joints with a chemical composition and microstructure similar to that of the BM appears to be efficient in maintaining its mechanical properties. In general, ferritic filler metals for joining 9%Ni steel have been under consideration for the last few decades as they would offer an economic benefit in terms of manufacturing costs. Matching filler metals for submerged arc welding (SAW) and gas tungsten arc welding (GTAW) have already been used in the fabrication of 9%Ni steel tubes [26] and in the construction of a laboratory-scale spherical model tank [27]. Recently, research on matching ferritic welding electrodes for shielded metal arc welding (SMAW) have shown attractive results, with an acceptable combination of tensile strength and impact toughness obtained [28]. Nevertheless, these filler metals are not yet extensively used for in-field welding, and more studies are still required in this important area, especially for the commonly used GMA welding process. The objective of the present work was to examine the feasibility of retaining the combination of mechanical properties for GMA-welded joints with 9%Ni steel using an experimentally fabricated matching ferritic filler metal with 11%Ni (ERNi11). For comparison, the commercially available and widely used Ni-based austenitic filler metal ERNiCrMo-3 (AWS-A5.14) was used.

## 2. Materials and Methods

The BM used was a ferritic 9%Ni steel grade ASTM A553 Type 1 (EN10028-4X8Ni9) with 14.5 mm plate thickness. The chemical composition and the mechanical–technological properties are given in Table 1.

Specimens with dimensions of 250 × 100 × 14.5 mm^3^ and a 60° double V-groove were used for GMA welding tests. This specimen size was defined on the basis of existing laboratory handling techniques. Prior practice with GMA welding tests has shown that a specimen length of at least 250 mm is enough to make a conclusion about weld seam quality. Double-sided butt-welded joints were made using the weld bead deposition sequence shown in Figure 1. Subsequently, preliminary tests were carried out. Based on these tests, the optimum welding parameters (Table 2) were selected. The aim was to obtain well-penetrated weld beads without fusion defects. In addition, undercut-free bead profiles were also a result. In this regard, the combination of 198 A welding current, 29 V arc voltage, and 210 mm/min–230 mm/min welding speed was used, which is in good agreement with well-established welding parameters. Welding was carried out for accurately assembled, aligned, mechanically clamped, and carefully restrained butt joints, as shown in Figure 2. Two types of 1.0 mm diameter filler metals were used, including the commercially available Ni-based austenitic filler metal ERNiCrMo-3 (AWS A5.14) and an experimentally produced matching ferritic filler metal with 11%Ni (ERNi11). Preheating and PWHT were applied to welds made using the matching ferritic filler metal ERNi11. Preheating was conducted at three temperature regimes: 150 °C (302 °F), 200 °C (392 °F), and 250 °C (482 °F). The preheating of the weld specimen was carried out with an oxy-acetylene flame. The preheating and interpass temperatures were monitored with a type K temperature sensor. The interpass temperature was maintained at 150 °C to minimize martensite formation and increase retained austenite in both the WM and HAZ of welded joints made using the ferritic filler metal and the HAZ of welded joints made using the austenitic filler metal. The specimen was welded as soon as the desired temperature was reached. PWHT was performed at a temperature of 580 °C (1076 °F). The holding time was 20 min [28]. The cooling down of samples was conducted in an ambient atmosphere.

After welding, a visual inspection of the GMA welds was performed. The occurrence of external weld defects such as undercuts was evaluated. The selected weld specimens were then subjected to X-ray testing (RT) to identify any internal defects. Samples for metallographic examinations were taken transversally to the welding direction. For metallurgical examinations, picric acid etchant was used for electrolytic etching of welded joints made using the Ni-based austenitic filler metal, while nital etchant was used for chemical etching of joints made using the matching ferritic filler metal. The geometric characteristics of the fusion zone were examined with a low magnification stereoscope, while the microstructures of the weld metal (WM), the material melted and re-solidified by the welding process, the heat-affected zone (HAZ), and the BM were investigated with an optical microscope. Comprehensive microscopic investigations including pattern quality, the phase map, the orientation color map, and grain size distribution were conducted using a scanning electron microscope equipped with an electron backscatter diffraction (EBSD) system. The EBSD, with a 2.53 µm step size, was used for the WM. Grains were detected using a grain detector angle of 3.92° and only grains larger than 100 pixels were considered. Compositional variations across the welded joints were determined using a scanning electron microscope equipped with an energy dispersive X-ray spectroscopy (EDS) unit at an accelerating voltage of 20 kV. Vickers microhardness measurements were performed using a Shimadzu 1000 g machine. The measurements were conducted on polished and etched cross sections at near mid-thickness of its WM, HAZ, and BM while applying a load of 500 g with a dwell time of 20 s. The tensile test was carried out at a constant traverse displacement rate of 2 mm/min using a Shimadzu 1000 kN hydraulic testing machine. Tensile specimens with a gauge length of 40 mm, a thickness of 14 mm, a width of 19 mm (in the range of the gauge length), a clamping range width of 25 mm, and a total length of 200 mm were machined in accordance with the ASME IX standard. The impact test was carried out at −196 °C (−320 °F) using a Roell Amsler 300 J pendulum impact tester. Standard impact test specimens, with the notch location in both the WM and the HAZ, were machined in accordance with the ASME IX standard.

For both tensile and impact tests, three specimens in the as-welded condition were tested for each welded joint. Impact tests were also conducted for both preheated and post-weld heat-treated joints. The average values of each property were considered and compared with those of the BM.

## 3. Results and Discussion

Visual inspection of GMA-welded joints made with the austenitic Ni base and matching ferritic filler metals revealed no noticeable external weld defects. RT confirmed fully penetrated welds with no unacceptable internal defects. This is mainly due to proper selection of the implemented welding conditions. The welds accepted after RT were sectioned and then metallographically and mechanically examined.

### 3.1. Metallurgical Examinations

Macro images of cross sections of welded joints made with the austenitic Ni-based filler metal and the matching ferritic filler metal are shown in Figure 3. It can be noticed that comparable fusion zone sizes were obtained due to the similar welding parameters applied to both joints. Weld development is essentially symmetrical about its center. Macroscopic observation confirmed the quality of the welds, which showed no internal defects (lack of fusion, porosity, cracks, etc.).

The microstructure of the base metal consists mainly of tempered martensite (Figure 4), and its average hardness value is 253 HV. The microstructure of the WM and the HAZ of a welded joint made using the nickel-based filler metal are shown in Figure 5. The width of the HAZ is 4 mm. Weld soundness was confirmed using a radiographic test and no internal welding defects were detected. The microstructure of the WM is a cast dendritic austenitic structure (Figure 5a), typical to that of the used austenitic filler metal [29]. The microstructure of the HAZ is a coarse-grained martensitic structure and the prior austenite grain boundaries can be seen (Figure 5b). This microstructure was formed due to high-temperature heating during welding followed by rapid cooling.

Optical microscopic photographs of the WM and the HAZ of a welded joint made using the matching ferritic filler metal are shown in Figure 6. Microscopic examination indicated that the width of the WM and HAZ is very close to that of the welded joint produced using the Ni-based austenitic filler metal. It also showed a sound WM where no unacceptable internal defects were found. The main observation is that the WM microstructure is completely different, consisting of a columnar cast dendritic martensitic structure typical for the ferritic filler metal used (Figure 6a). The coarse-grained HAZ microstructure (Figure 6b) is fairly comparable to that of the welded joint produced using the austenitic filler metal (Figure 5b), where a martensitic structure with a small amount of retained austenite was found. This is due to similar heat inputs in both cases. The microstructure of the HAZ bordering the BM exhibited a less martensitic structure due to the lower heating temperature.

SEM with an EBSD detector was used to study the microstructure in more detail. The pattern quality, phase map, EBSD orientation color map, and grain size distribution of the Ni-based WM are shown in Figure 7. A columnar dendritic structure was obtained for the austenitic weld metal (Figure 7a). The phase map, Figure 7b, confirmed the fully austenitic microstructure. A coarse austenitic structure with high-angle grain boundaries is seen in the orientation color map (Figure 7c). The weld metal is characterized by a significant variation in grain size, as shown by the histogram of grain size distribution. The number of grains detected was 274, the smallest grain size was 10 μm, the largest was 910 μm, and the average grain size was 582 µm (Figure 7d).

For the matching ferritic weld metal, Figure 8 shows the pattern quality, the phase map, the EBSD orientation color map, and the grain size distribution. The coarse columnar dendritic martensitic structure of this ferritic weld metal is confirmed by the pattern quality shown in Figure 8a. The martensitic microstructure and the retained austenite constitute 98.4% and 1.6%, respectively (Figure 8b).

Published data [17] on EBSD analyses of a 9%Ni steel’s BM showed a microstructure of fine tempered martensite with a 10 µm grain size and a small amount of retained austenite. In contrast to the BM, the orientation color map of the WM provided in Figure 8c shows a coarser martensite structure with small-angle grain boundaries. Significant variation in grain size was detected in the weld by the grain size distribution histogram, with the smallest grain size being 5 μm and the largest 175 μm; the average grain diameter was 42.6 µm, which is much coarser than in the BM, and the total number of examined grains was 11,825 (Figure 8c).

Since the weld metal in GMAW is actually a mixed material of BM and filler wire, the dilution rate (DR) of the BM in WM was estimated for the first pass. Due to the double V-groove geometry, the first pass is expected to have the highest DR of the filler passes. With the welding parameters used (Table 2), the wire feed rates in the welding tests were 10 m/min. Since the welding tests were carried out with a solid wire of 1 mm diameter, the area of the melted welding wire was about 7.85 mm^2^ per pass. From the macrographs in Figure 4, it could be determined planimetrically that the area of the first pass was about 13 mm^2^. Of this area, 5.15 mm^2^ or 40.0% is accounted for by the diluted BM.

### 3.2. Hardness Measurements

Figure 9 shows representative hardness distributions across welded joints made using the austenitic Ni base and the matching ferritic filler metals, as well as those of the 200 °C preheated joint. The highest variation in the hardness measurements was HV 0.5 ± 3%. Adequate weld zone and HAZ widths of 15.5 mm and 4.0 mm were obtained for all joints. The hardness profiles of the welded joints are significantly influenced by the filler metal type. The Ni-based austenitic filler metal resulted in a weld hardness value of 235 HV, which is close to the hardness of the BM (253 HV). The matching ferritic filler metal resulted in WM with a higher hardness (~345 HV), a result of its martensitic structure. This hard structure is expected to maintain the welded joint’s tensile strength while negatively affecting its impact toughness. The hardness values of both the WM and HAZ of the 200 °C preheated joint were significantly decreased (~300 HV) due to its less martensitic structure and an increased fraction of retained austenite. It should be reported that lower preheating temperature (150 °C) resulted in a more martensitic structure due to a higher cooling rate that in turn maintained higher hardness values, which leads to an adverse effect on impact toughness. On the other hand, the 250 °C preheated joint showed a hardness profile similar to that of the 200 °C preheated joint. As a result, 200 °C was considered to be the optimum preheating temperature since it also has a positive effect on the manufacturing cost. It is also important to highlight that PWHT resulted in a hardness profile close to that of the optimum preheated joint and similar to that of previously published research on SMAW with 9%Ni steel [28]. This is due to a tempered martensitic structure with a more stable retained austenite.

The implemented PWHT temperature of 580 °C was decided based on a well-settled and accepted range for 9%Ni steel. PWHT above this temperature can result in a loss of toughness due to unstable temper-austenite, which transforms to martensite at subzero temperatures [30].

### 3.3. Micro Chemical Analysis

EDS microanalysis was used to study the variations in chemical composition along the HAZ and WM in two specimens, one welded with an austenitic Ni-based filler metal and the other one welded with a matching ferritic filler metal (Figure 10 and Figure 11). For the joint welded with the Ni-based austenitic filler metal, ERNiCrMo-3, the chemical composition of the weld is completely different from that of the BM. Nickel content and chromium content are much higher in the WM compared to the BM, while the iron content in the weld is much lower than that of the 9%Ni BM. On the other hand, the composition of the WM is similar to that of the BM in the case of joints welded with the matching filler metal.

Regarding welded joints made using the matching ferritic filler metal, no noticeable variation in the chemical composition of its WM or BM was found (Figure 11). In other words, a weld with a chemical composition similar to that of the BM was obtained that in turn resulted in a hard weld metal due to its martensitic structure. Concerning this, higher tensile strength and lower fracture toughness are expected for welded joints made using the matching ferritic filler metal in comparison to those produced using the Ni-based austenitic filler metal.

### 3.4. Mechanical Properties of Welded Joints

Figure 12 shows pictures of tensile test fracture specimens from welded joints made with austenitic Ni-based and matching ferritic filler metals. The fracture occurred in the deposited weld for the austenitic Ni-based filler metal (Figure 12a), while it occurred in the base metal of the specimen welded with the matching ferritic filler metal (Figure 12b). Concerning this, the ruptures took place in the reduced tensile strength zone of both joints. Unlike expected, welded joints made using the Ni-based austenitic filler metal showed relatively low elongation. This may be explained by a higher tensile strength difference between its weld and the base metal, as well as by a larger difference in the thermal expansion coefficients of its weld and its BM, which led to welding residual stresses and lower overall elongation.

The tensile strength of the BM is equal to 745 MPa, the joints welded with the austenitic nickel-based filler metal (ERNiCrMo-3) exhibited tensile strength of 685 MPa, while welding with a similar filler metal (ER11Ni) resulted in tensile strength of 749 MPa (Figure 13). Thus, welding 9%Ni steel with similar ferritic filler metal resulted in higher strength comparable to that of the BM. This can be attributed to the similar microstructures of the WM and BM, which are composed of hard martensite and soft retained austenite.

The weld metal’s microstructure is seen as the most important parameter for retaining the tensile strength of the welded joint. Unlike the joints welded with austenitic Ni-based filler metal, the tensile strength of the welded joints with ferritic filler metal complies with the ASME code, which specifies that the minimum tensile strength of the welded joint should be equal to that of the BM. SEM photographs of the fracture surfaces of specimens obtained from welded joints made with the austenitic Ni-base filler metal and the matching ferritic filler metal are shown in Figure 14. Dimples were observed in both specimens, indicating a ductile fracture mode. This is because fractures occurred in the softer Ni-based austenitic WM (Figure 14a) and in the tempered martensitic BM (Figure 14b).

As for the impact test, a V-notch was cut at the center of the WM and the HAZ using a high precision CNC milling machine to determine the impact toughness of both zones (Figure 15a,b). Figure 16 illustrates the impact test results for the conditions investigated in this study for tests conducted at −196 °C. BM showed the highest impact toughness of 178 J. This can be attributed to its microstructure of tempered martensite and retained austenite. The impact toughness of the HAZ was equal to 66 J and 68 J for joints made using the ferritic filler metal and the Ni-based austenitic filler metal, respectively. This is attributed to a similar microstructure and the heating–cooling cycles in both cases. The impact toughness of the WM produced using a similar ferritic filler metal (ERNi11) is equal to 71 J due to its martensitic structure. This value is still well above the ASME minimum specified value of ≥34 J.

Applying preheating or post-weld heat treatment on welds produced using ERNi11 decreased the mismatching of toughness between the WM and HAZ; in this case, the absorbed energy was in the range of 107 J to 119 J, as seen in Figure 16.

Figure 17 shows SEM photographs of the impact fracture surface of joints produced with the Ni-based austenitic and the ferritic filler metals. Compared to the WM fracture surface of joints produced using the Ni-based austenitic filler metal (Figure 17a), a less ductile fracture appearance was obtained for joints produced using the matching ferritic filler metal where dimples were not clearly visible (Figure 17b). On the other hand, the impact fracture surface of both preheated and post-weld heat-treated joints showed a ductile fracture surface. SEM photographs of the impact fracture surface of 200 °C preheated and PWHT joints made using the matching ferritic filler metal are shown in Figure 18. The most important thing to note is the clear visible dimples for the fracture surface of the 200 °C preheated joint (Figure 18a), indicating a ductile fracture surface. This is due to the less martensitic structure with an increased fraction of retained austenite. This result is comparable to that of a previous study on laser beam welding with the same material [31]. A ductile impact fracture surface was also obtained for the post-weld heat-treated joint (Figure 18b) due to its WM’s tempered martensitic structure with more stable retained austenite. The effect of PWHT is in agreement with a previous study on SMAW with 9%Ni steel [29]. Tensile and impact properties are influenced not only by the microstructure of the fusion zone, but also by the degree of its mechanical mismatching depending on the type of the filler metal. A welded joint with a homogeneous microstructure resulting in lower mechanical mismatching is very important for obtaining an adequate combination of tensile strength and impact toughness. This is achievable with the low-cost matching ferritic filler metal.

## 4. Conclusions

Compared to welded joints made using the austenitic Ni-based filler metal ERNiCrMo-3 (685 MPa), a higher tensile strength (749 MPa) similar to that of the base metal (745 MPa) was obtained using the ferritic filler metal due to its weld metal’s martensitic structure. However, this WM’s hard martensitic structure resulted in low impact toughness (71 J); however, this is still more than two times higher than the minimum specified value (≥34 J). The most important thing to notice is the lower mismatching for both the tensile strength and impact toughness of the WM and HAZ of this welded joint due to its similar microstructure. The overall tensile and impact properties of the welded joint are affected by its microstructure, as well as its mechanical mismatching as a function of the filler metal. Welded joints with lower mechanical mismatching are important for obtaining an adequate combination of mechanical properties. This is viable using the lower cost matching ferritic filler metal. For further desirable combinations of mechanical properties, either preheating or PWHT should be applied. Preheating results in a less martensitic structure with an increased fraction of retained austenite. PWHT results in a tempered martensitic structure with more stable retained austenite. However, a low preheating temperature of 200 °C is an easier, more applicable, and cost-effective solution compared to the high PWHT temperature of 580 °C. It should be noted that the preheating of large components such as cryogenic vessels can be very time consuming and expensive. Therefore, the use of mobile inductive preheating systems should be considered, e.g., an inductor that runs along with the welding process and heats the part locally. Such systems are already available on the market.

## Figures and Tables

**Figure 1 materials-15-08538-f001:**
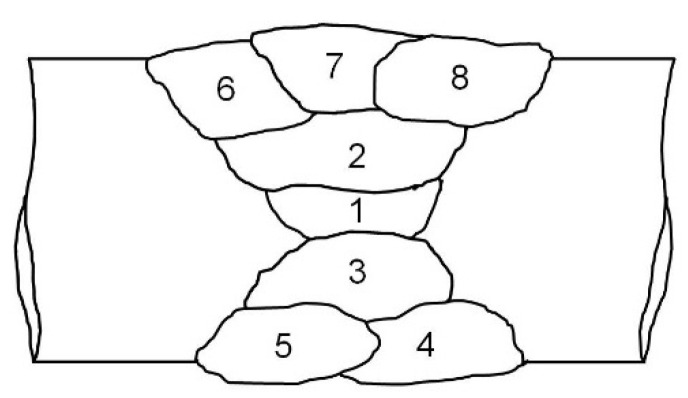
Schematic illustration of the weld bead deposition sequence for two-sided (60° double V-groove) GMA butt-welded joints.

**Figure 2 materials-15-08538-f002:**
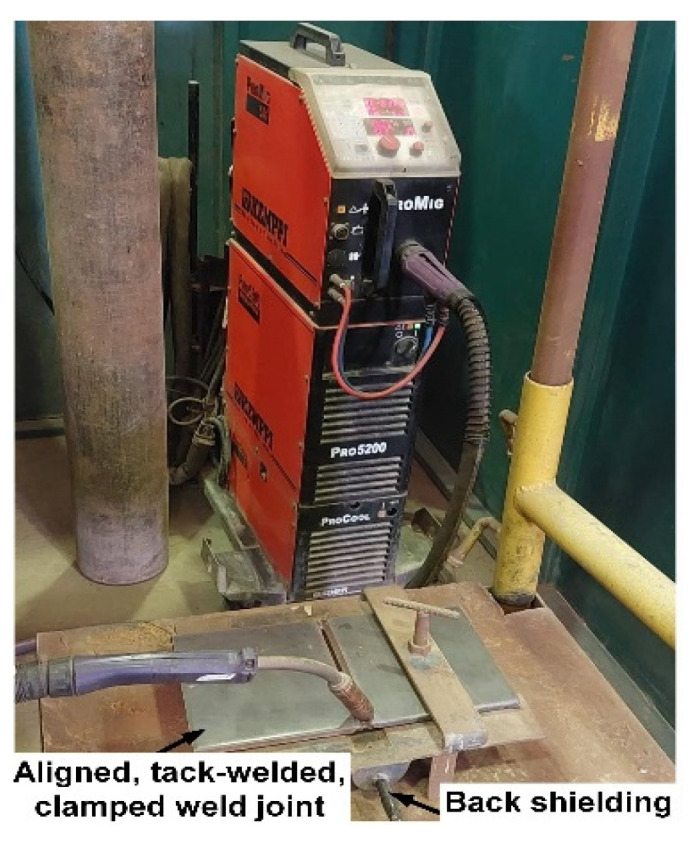
Experimental setup of GMA welding experiments.

**Figure 3 materials-15-08538-f003:**
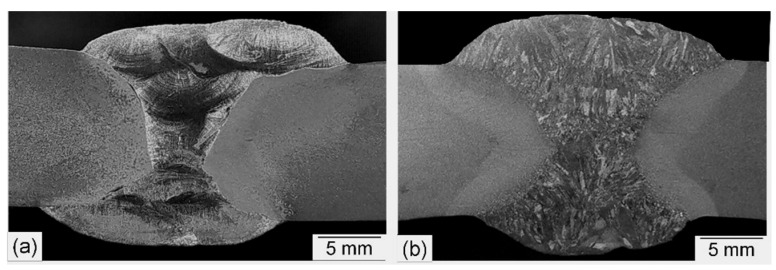
Macro images of cross sections of welded joints produced using the Ni-based austenitic (**a**) and the matching ferritic (**b**) filler metals.

**Figure 4 materials-15-08538-f004:**
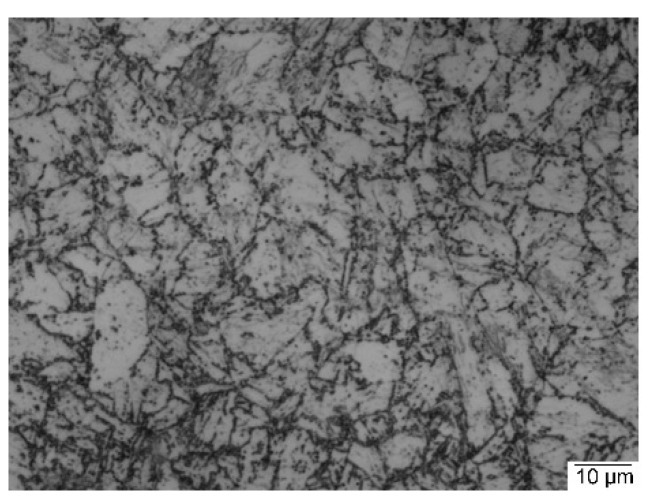
Optical microscopic photograph of the used 9%Ni steel’s BM.

**Figure 5 materials-15-08538-f005:**
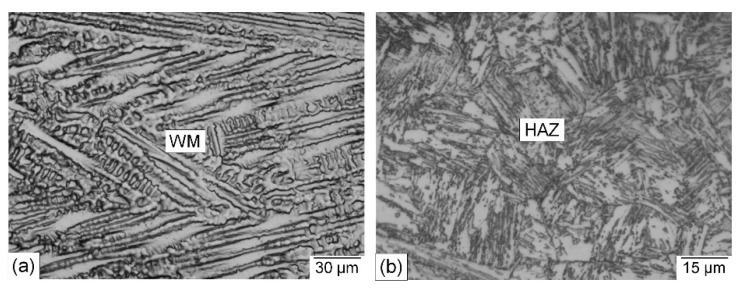
Optical microscopic photographs of a cross section of a welded joint produced using the Ni-based austenitic filler metal: (**a**) WM; (**b**) HAZ.

**Figure 6 materials-15-08538-f006:**
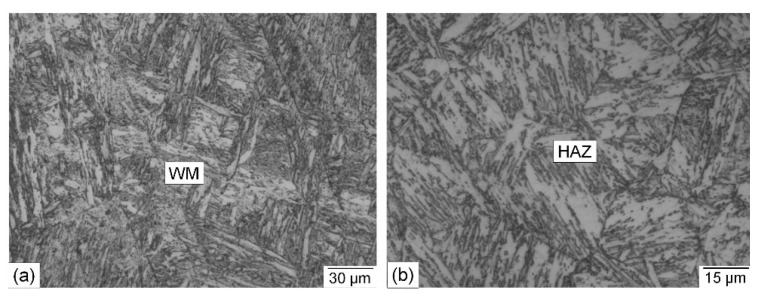
Optical microscopic photographs of a cross section of a welded joint made using the matching ferritic filler metal: (**a**) WM; (**b**) HAZ.

**Figure 7 materials-15-08538-f007:**
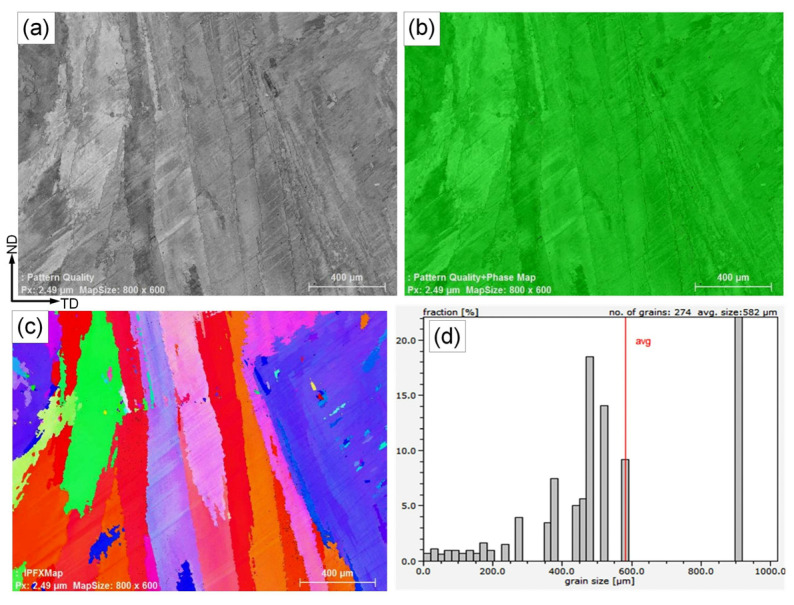
Pattern quality (**a**), phase map (**b**), EBSD orientation color map (**c**), and grain size distribution histogram (**d**) of the weld metal deposited using the Ni-based austenitic filler metal.

**Figure 8 materials-15-08538-f008:**
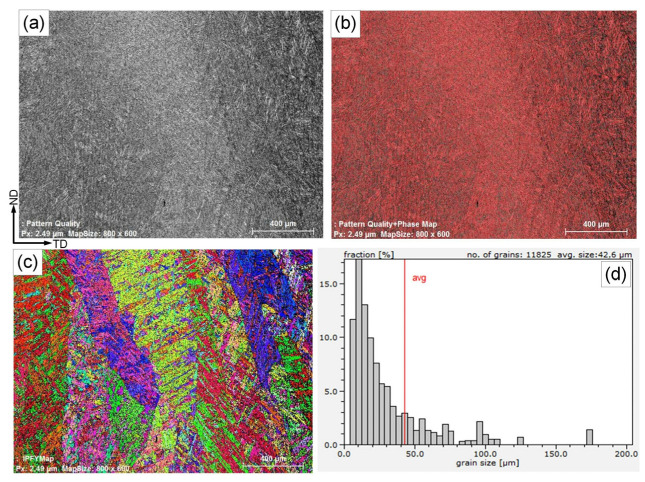
Pattern quality (**a**), phase map (**b**), EBSD orientation color map (**c**), and grain size distribution histogram (**d**) of the weld metal deposited using the matching ferritic filler metal.

**Figure 9 materials-15-08538-f009:**
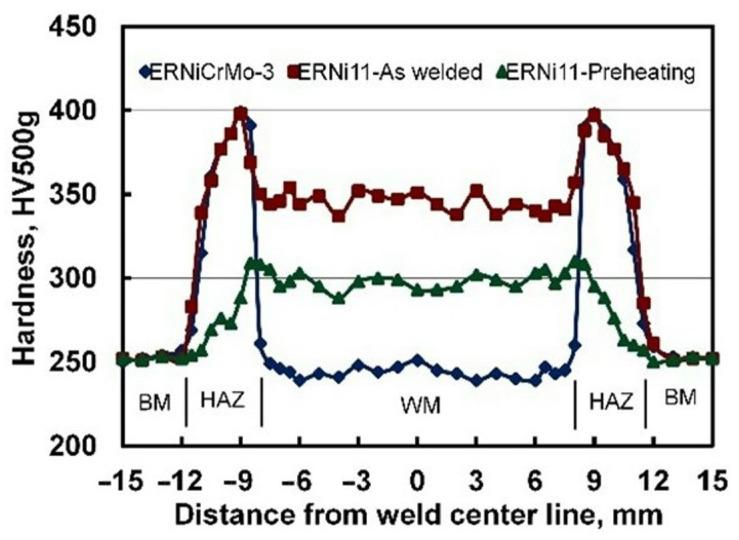
Hardness profiles of as-welded joints produced using the Ni-based austenitic and the matching ferritic filler metals together with that of the 200 °C preheated joint.

**Figure 10 materials-15-08538-f010:**
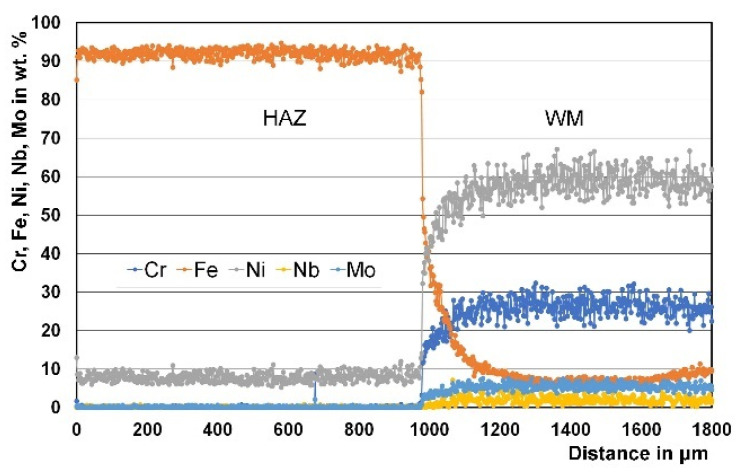
Example of EDS line scan microanalysis through the HAZ and WM deposited using the Ni-based austenitic filler metal.

**Figure 11 materials-15-08538-f011:**
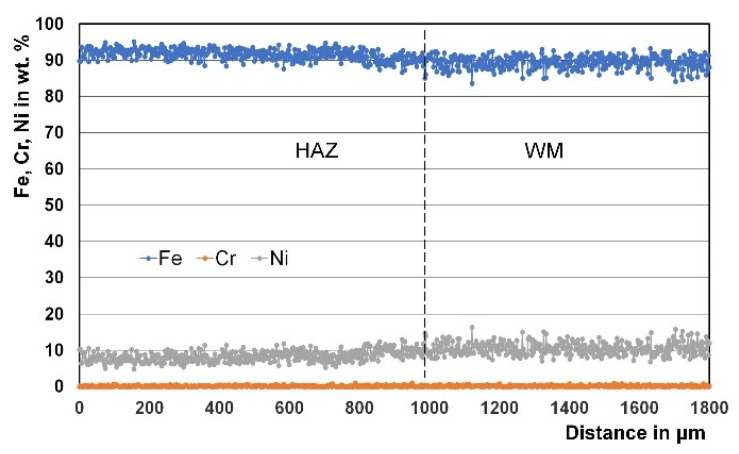
Example of EDS line scan microanalysis through the HAZ and WM deposited using the matching ferritic filler metal.

**Figure 12 materials-15-08538-f012:**
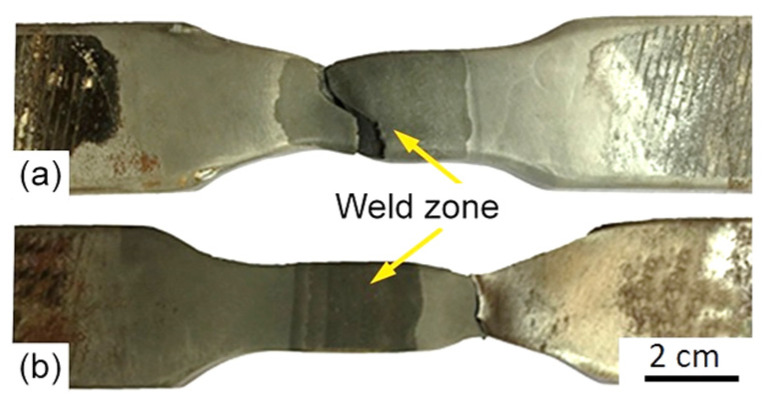
Photographs of tensile fracture specimens from joints produced with the austenitic (**a**) and the ferritic (**b**) filler metals.

**Figure 13 materials-15-08538-f013:**
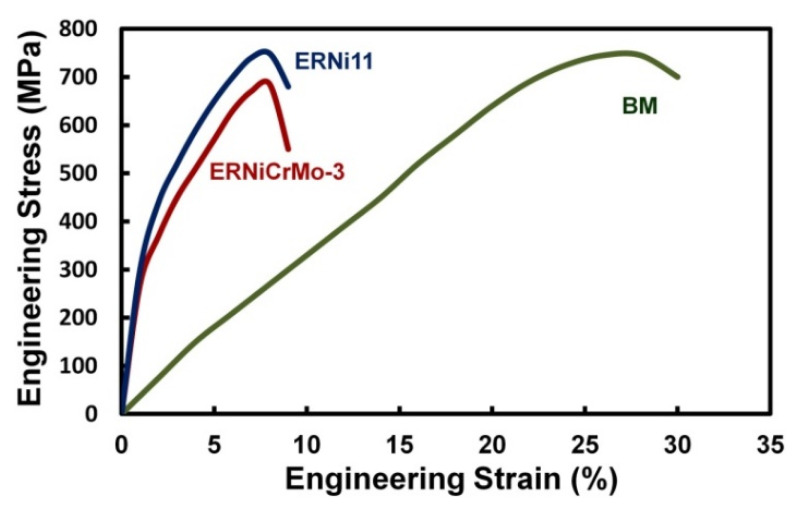
Engineering stress–strain curves of joints produced with the Ni-based austenitic (ERNiCrMo-3) and the matching ferritic (ERNi11) filler metals, as well as the BM.

**Figure 14 materials-15-08538-f014:**
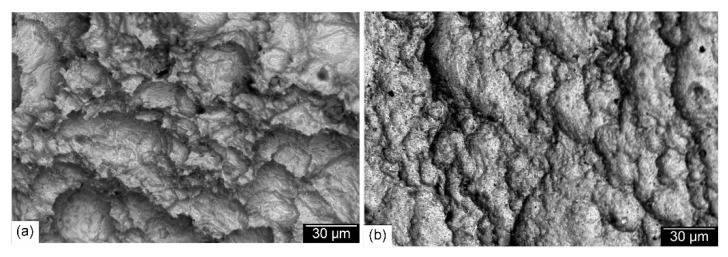
SEM photographs of the tensile fracture surfaces of joints produced with the Ni-based austenitic (**a**) and the matching ferritic (**b**) filler metals.

**Figure 15 materials-15-08538-f015:**
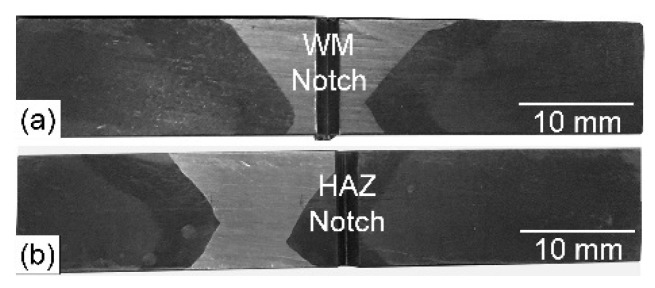
Photographs of V-notch impact test specimens with a notch location in the WM (**a**) and in the HAZ (**b**).

**Figure 16 materials-15-08538-f016:**
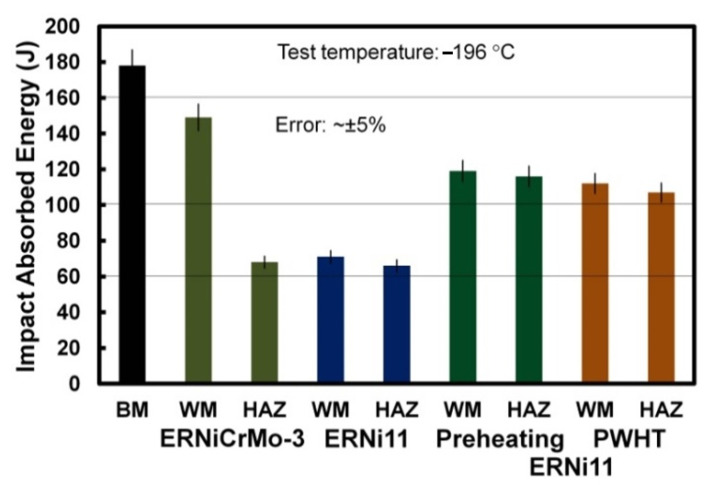
Impact absorbed energy for the WM and HAZ of joints produced with ERNiCrMo-3 and ERNi11 filler metals, together with that of the 200 °C preheated joints, PWHT joints, and the BM.

**Figure 17 materials-15-08538-f017:**
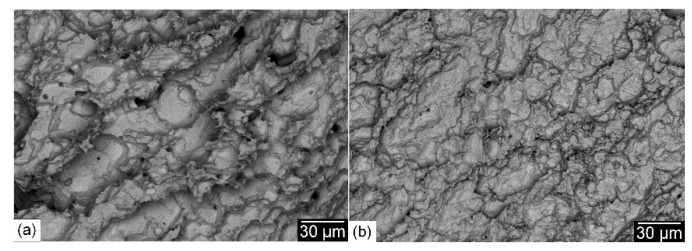
SEM photographs of the impact fracture surface of joints produced with the Ni-based austenitic (**a**) and the ferritic (**b**) filler metals.

**Figure 18 materials-15-08538-f018:**
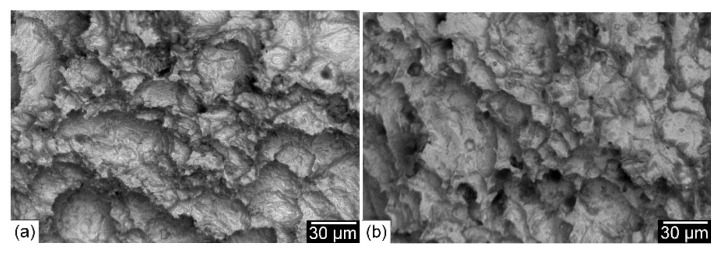
SEM photographs of the impact fracture surface of 200 °C preheated (**a**) and PWHT (**b**) joints produced with the ferritic filler metal.

**Table 1 materials-15-08538-t001:** Chemical composition (ASTM E-3) and mechanical properties (ASTM A370/ASTM D6110/ASTM E384) of the 9%Ni steel used.

Chemical Composition (wt.%)									
C	Si	Mn	P	S	Al	Ni	Cr	Cu	Fe
0.07	0.20	0.57	0.003	0.001	0.02	9.15	0.03	0.02	balance
Mechanical properties									
Yield strength (MPa)		Tensile strength (MPa)		Elongation (%)		Hardness (HV)		Impact absorbed energy at −196 °C (−320 °F)	
671		745		28		253		178	

**Table 2 materials-15-08538-t002:** Optimum welding parameters used for two-sided GMA butt-welded joints.

Pass No.	Current (A)	Voltage (V)	Speed (mm/min)	HI (kJ/mm)
1	198	29	222	1.55
2	198	29	210	1.64
3	198	29	210	1.64
4	198	29	224	1.54
5	198	29	224	1.54
6	198	29	230	1.50
7	198	29	230	1.50
8	198	29	230	1.50

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
