# Peer review of "Retaining Mechanical Properties of GMA-Welded Joints of 9%Ni Steel Using Experimentally Produced Matching Ferritic Filler Metal"

_materials, 2022, doi:10.3390/ma15238538_

Round 1
Reviewer 1 Report
This paper has studied the feasibility of using matching ferritic filler metal, ERNi 11 for welding 9%Ni steel to replace the currently used Ni-based austenitic filler metal, ERNiCrMo-3, and demonstrated that it is feasible to use an experimentally produced matching ferritic filler metal to maintain tensile strength. The authors have completed a very meaningful work。
Author Response
Thank you very much and sincere appreciation for your valuable time and great efforts in reviewing the manuscript.
Reviewer 2 Report
Authors reported the retaining mechanical properties of GMA welded joints of 2 9%Ni steel using experimentally produced matching ferritic filler metal. This paper can be accepted after minor revision.
(1) Please include the improvement in tensile strength of the joints in abstract.
(2) Provide the most important and focused keywords alone.
(3) What is the novelty of the work ? include the novelty of the work at the end of the introduction section.
(4) Table 1. Chemical composition and mechanical properties of used 9%Ni steel. – how these values are obtained? Have authors tested?
(5) Please specify the ASTM standard followed for all the tests.
(6) The error bar is not provided properly in all the bar charts. Please specify the values along with the error.
(7) Stress train curves for tensile testing is missing.
(8) Why aim of the work is provided in conclusion section?
(9) Revise the conclusion section with numerical support and percentage improvement in properties.
(10)No references are found form the year 2022 and only one reference form 2021. Please cite the recent references.
(11)Please improve the discussion with literatures support and mention the reasons for changes occurred in the microstructure and properties.
Author Response
Thank you very much for your valuable and important additions. All pointed out recommendations / comments have been considered as following:
(1) Please include the improvement in tensile strength of the joints in abstract.
Answer:
Improvement in tensile strength of the joints has been included in abstract.
(2) Provide the most important and focused keywords alone.
Answer:
Keywords have been shortened to include only the most important and focused
keywords (9 Keywords were reduced to 7 Keywords)
(3) What is the novelty of the work? include the novelty of the work at the end of the introduction section.
Answer:
The effect of PWHT on impact properties of SMA welded joint of 9%Ni steel has been previously clarified in published research papers. The current manuscript is concerned with GMA welding process, which is much more widely used in actual application for manufacturing large size LNG transportation and storage facilities, in comparison with other welding processes including SMAW as well as TIG welding processes.
In the current manuscript and for comparison purpose, the effect of preheating on impact properties was also studied. The novelty of this work is that low preheating temperature (200 °C) is much more manageable, practicable and economical solution for improving impact properties of welded joint, in comparison with PWHT high temperature (580 °C) of previously published works on SMAW.
The novelty was added at the end of abstract section.
(4) Table 1. Chemical composition and mechanical properties of used 9%Ni steel. – how these values are obtained? Have authors tested?
Answer:
Yes, chemical composition and mechanical properties given in Table 1 were obtained based on actually conducted analysis/tests for this research work.
(5) Please specify the ASTM standard followed for all the tests.
Answer:
- ASTM standards followed for different tests of the base metal were specified and added in Table 1. ASTM E-3 for chemical analysis, ASTM A370 for tensile test, ASTM D6110 for Charpy impact test, ASTM E384 for Vickers hardness test.
- As for tensile test, impact test and hardness test of welded joints, ASME IX standard was followed since this standard is concerned with weldments of pressure vessels.
(6) The error bar is not provided properly in all the bar charts. Please specify the values along with the error.
Answer:
The error bar values are specified and added in all the bar charts (Figure 14 & Figure 17).
(7) Stress strain curves for tensile testing are missing.
Answer:
Stress strain curves were obtained as photos (not excel files) based on the available machine capability. It was difficult to include tensile test results in various individual/separate curves. Then, tensile test results were presented in bar chart.
(8) Why aim of the work is provided in conclusion section?
Answer:
Many thanks for pointing out such mistake. Aim of the work was removed from conclusion section.
(9) Revise the conclusion section with numerical support and percentage improvement in properties.
Answer:
The conclusion section was revised where numerical support and percentage improvement in properties were added.
(10) No references are found form the year 2022 and only one reference form 2021. Please cite the recent references.
Answer:
One more recent reference from 2021 (Ref No. 29) was cited.
To the best of our knowledge, no references from the year 2022 were found in the current research area.
(11) Please improve the discussion with literatures support and mention the reasons for changes occurred in the microstructure and properties.
Answer:
This valuable and important recommendation has been carefully considered for improving the discussion section. The reasons for changes occurred in the microstructure and properties were elaborated.
Reviewer 3 Report
Dear Authors, the paper is interesting and in general well structured.
Some sugestions:
Introduction:
This sentence, considering the actual international scenario, probably need to be actualizzed: "The global need to use natural gas as a clean energy source with lower CO2 emissions 36 than petroleum is increasing to reduce the problem of environmental contamination. Be-37 sides, the natural gas lower price has a remarkable positive economic impact in different 38 industrial sectors including mainly fertilizer, chemical and power plants"
Today, in EUROPE the price of the NG is very HIGH, sure the link to the big gassificators is even more ease to do and increase the interest inthe readers considering that the new experimental ferritic filler wire should be cheap than high Ni alloy filler materials.
Check the Acronims; I.e. WM is used before you mention it for the first time, I sugest you to explain what intend for Welded Metal.
In your paper is missing a discussion about the diluition ratio ( DR) of BM in Fusion Zone ( Weld Metal) sometimes in fact Weld Metal is attribuite to all FIller material deposited but really in welded Joint you have a mixed material (BM+ Filler wire) this acspect need to be considered in particula in first pass where the DR is highere than the successive passes.
What about the interpass Temperature? The argument is missing please add some information about than toghether with HI and Pre heat control the microstructure of the joint.
You mention X ray NDT but there are not discussion or pictures about;
You use a not ISO standar welding aulification Coupons 250 x 100 mm, usualluy ISO stnadard and I think ASME too, specify coupon with minimum dimension 350 x 150 mm, NO problem but I sugest to add some sentence that justify the use of redused dimensions welded coupons becouse it is a preliminary research work. This is valid to justify the missing of bending test that you know is a more valuable tes of ductility of the welded Joint.
About the result of tensile test, your consideration are correct, and correctly you do not report data about Yeld Strenght. At the same mode even the data about elongation are just for speaking becouse the test is a transversal tensile test and the condiserations about elongation are not real becous yu test 3 in series materials ( BM, HAZ, WM,HAZ, BM).
I think that even the bending test will be performed and demostrate the ductiliy of the weld the development of filler materials paret of BM is very interesting.
Need to be evaluate the cost of preheting on very big componets like cryogenic vessels.
Author Response
Dear Reviewer, thank you very much for your valuable and important additions. All pointed out recommendations and comments have been considered as following:
1. Introduction:
This sentence, considering the actual international scenario, probably need to be actualizzed: "The global need to use natural gas as a clean energy source with lower CO2 emissions than petroleum is increasing to reduce the problem of environmental contamination. Besides, the natural gas lower price has a remarkable positive economic impact in different industrial sectors including mainly fertilizer, chemical and power plants"
Today, in EUROPE the price of the NG is very HIGH, sure the link to the big gassificators is even more ease to do and increase the interest inthe readers considering that the new experimental ferritic filler wire should be cheap than high Ni alloy filler materials.
Answer: I fully agree with you. The sentence was modified.
2. Check the Acronims; I.e. WM is used before you mention it for the first time, I sugest you to explain what intend for Welded Metal.
Answer: yes, the correction is done on the page 4.
3. In your paper is missing a discussion about the diluition ratio (DR) of BM in Fusion Zone (Weld Metal) sometimes in fact Weld Metal is attribuite to all FIller material deposited but really in welded Joint you have a mixed material (BM+ Filler wire) this acspect need to be considered in particula in first pass where the DR is highere than the successive passes.
Answer: thank you for this valuable comment. The estimation of the delution rate is added in the discussion section
4. What about the interpass Temperature? The argument is missing please add some information about than toghether with HI and Pre heat control the microstructure of the joint.
Answer: yes, the interpass temperature was controlled. Corresponding data have been added in the "Materials and methods" section.
5. You mention X ray NDT but there are not discussion or pictures about;
Answer: You are right. We mentioned X-Ray tests because it was on the basis of these tests that the samples were selected for further investigation. Since the X-Ray tests were performed only exemplarily on selected samples, we decided not to include the X-Ray results in the text. At this stage of the research, the statistics on the internal weld quality are still needed. In this context, further investigations are still necessary.
6. You use a not ISO standar welding aulification Coupons 250 x 100 mm, usualluy ISO stnadard and I think ASME too, specify coupon with minimum dimension 350 x 150 mm, NO problem but I sugest to add some sentence that justify the use of redused dimensions welded coupons becouse it is a preliminary research work. This is valid to justify the missing of bending test that you know is a more valuable tes of ductility of the welded Joint.
Answer: I completely agree with this comment. We use subsamples for preliminary research due to the available laboratory capacity. An explanation is given in the text.
7. About the result of tensile test, your consideration are correct, and correctly you do not report data about Yeld Strenght. At the same mode even the data about elongation are just for speaking becouse the test is a transversal tensile test and the condiserations about elongation are not real becous yu test 3 in series materials ( BM, HAZ, WM,HAZ, BM).
Answer: Yes, I share with you the same point of view regarding tensile testing of welded specimens.
8. I think that even the bending test will be performed and demostrate the ductiliy of the weld the development of filler materials paret of BM is very interesting.
Answer: yes, i am agree with you, the bend test is quite significant. Unfortunately, we did not have enough sample material to perform the bend test. However, we will gladly take your recommendation into account for further research.
9. Need to be evaluate the cost of preheting on very big componets like cryogenic vessels.
Answer: That's a fair comment. Preheating a whole large volume component can be really time consuming and expensive. Therefore, we recommend using mobile systems for inductive preheating, e.g. an inductor that runs along with the welding process and heats the part locally. Such systems are already available on the market. A corresponding remark is given in the text.
Reviewer 4 Report
The study submitted for review deals with a topical and important aspect of welding LNG transport tanks.
The content of the study as well as the descriptions included and the interpretation of the results are not objectionable. However, several issues need to be corrected or clarified:
1) Section 3.1 refers to the visual inspection of the external made connections. I suggest including photographs of the connections made, which will allow the reader to objectively refer to the description provided.
2) Fig. 4a - do the authors have a photo of a "better" weld cross section. The cross-section on the picture is not very symmetrical which spoils the overall perception of the picture.
3) The description of the microhardness measurements from line 243 onwards refers to information that is not shown on the graph. Why are the microhardness profiles not included for the described cases of sample heating processes?
4)Fig. 17 - Poorly legible summary of impact measurement results. The captions for the individual bars are unclear. I suggest introducing colours for the individual bars in the graph and a description in the form of a graph legend. Additionally the graph should be centred.
Author Response
Many thanks and appreciation for your valuable and important pointed out issues. All pointed out recommendations / comments have been considered as following:
(1) Section 3.1 refers to the visual inspection of the external made connections. I suggest including photographs of the connections made, which will allow the reader to objectively refer to the description provided.
Answer:
Visual inspection is a preliminary method to check external defects of welded joints. Macroscopic examination of cross sections of welded joints (Figure 4) is more effective and accurate method to check both external and internal welding defects. Therefore, photograph of the connections made was not included.
(2) Fig. 4a - do the authors have a photo of a "better" weld cross section. The cross-section on the picture is not very symmetrical which spoils the overall perception of the picture.
Answer:
Unfortunately, we don’t have a better photo for Fig. 4a. Yes, the cross section of Fig. 4a is not very symmetrical. It may be still acceptable since such degree of non-symmetrical has almost no negative effect on chemical, metallurgical, mechanical properties.
(3) The description of the microhardness measurements from line 243 onwards refers to information that is not shown on the graph. Why are the microhardness profiles not included for the described cases of sample heating processes?
Answer:
It was difficult to include many hardness profiles (6 profiles) in the same Figure:
- As-welded joint made using the Ni-based austenitic filer metal;
- As-welded joint made using the matching ferritic filler metal;
- 150 oC preheated joint;
- 200 oC preheated joint;
- 250 oC preheated joint;
- Post weld heat treated joint (580 oC for 20 minutes).
The reason for that is to minimize overlapping of different profiles for clear comparison, particularly half of these profiles were almost similar/close to each other (200 oC preheated joint, 250 oC preheated joint, post weld heat treated joint.
Therefore, only the following 3 profiles were combined in one Figure for clear/feasible comparison:
- As-welded joint made using the Ni-based austenitic filer metal;
- As-welded joint made using the matching ferritic filler metal;
- 200 oC preheated joint, as an optimum preheating temperature.
(4) Fig. 17 - Poorly legible summary of impact measurement results. The captions for the individual bars are unclear. I suggest introducing colours for the individual bars in the graph and a description in the form of a graph legend. Additionally the graph should be centered.
Answer:
The individual bars for different impact results have been demonstrated in colors for clear/visible comparison. And the graph was adjusted in the center.